# Knowledge Distillation from Few Samples

## Abstract

Current knowledge distillation methods require full training data to distill knowledge from a large "teacher" network to a compact "student" network by matching certain statistics between "teacher" and "student" such as softmax outputs and feature responses. This is not only time-consuming but also inconsistent with human cognition in which children can learn knowledge from adults with few examples. This paper proposes a novel and simple method for knowledge distillation from few samples. Taking the assumption that both "teacher" and "student" have the same feature map sizes at each corresponding block, we add a $1 \times 1$ conv-layer at the end of each block in the student-net, and align the block-level outputs between "teacher" and "student" by estimating the parameters of the added layer with limited samples. We prove that the added layer can be absorbed/merged into the previous conv-layer to formulate a new conv-layer with the same size of parameters and computation cost as the previous one. Experiments verify that the proposed method is very efficient and effective to distill knowledge from teacher-net to student-net constructing in different ways on various datasets.

## 1 Introduction

Deep neural networks (DNNs) have demonstrated extraordinary success in a variety of fields such as computer vision (Krizhevsky & Hinton, 2012; He et al., 2016), speech recognition (Hinton et al., 2012), and natural language processing (Mikolov et al., 2010). However, DNNs are resource-hungry which hinders their wide deployment to some resource-limited scenarios, especially low-power embedded devices in the emerging Internet-of-Things (IoT) domain. To address this limitation, extensive works have been done to accelerate or compress deep neural networks. Putting those works on designing (Chollet, 2016) or automatically searching efficient network architecture aside (Zoph & Le, 2016), most studies try to optimize DNNs from four perspectives: network pruning (Han et al., 2016; Li et al., 2016), network decomposition (Denton et al., 2014; Jaderberg et al., 2014), network quantization (or low-precision networks) (Gupta et al., 2015; Courbariaux et al., 2016; Rastegari et al., 2016) and knowledge distillation (Hinton et al., 2015; Romero et al., 2015).

Among these method categories, knowledge distillation is somewhat different due to the utilization of information from the pre-trained teacher-net. The concept was proposed by (Bucila et al., 2006; Ba & Caruana, 2014; Hinton et al., 2015) for transferring knowledge from a large "teacher" model to a compact yet efficient "student" model by matching certain statistics between "teacher" and "student". Further research introduced various kinds of matching mechanisms in the field of DNN optimization. The distillation procedure designs a loss function based on the matching mechanisms and enforces the loss during a full training process. Hence, all these methods usually require time-consuming training procedure along with fully annotated large-scale training dataset.

Meanwhile, some network pruning (Li et al., 2016; Liu et al., 2017) and decomposition (Zhang et al., 2016; Kim et al., 2016) methods can produce extremely small networks, but with large accuracy drops so that time-consuming fine-tuning is required for possible accuracy recovery. Usually, it may still not be able to recover the accuracy drops with the original cross-entropy loss due to its low representation capacity. Hence, knowledge distillation may be used to alleviate the problem, since the compact student-net can sometimes be trained to match the performance of the teacher-net. For instance, Crowley et al. (2017) uses cheap group convolutions and pointwise convolutions to build a small student-net and adopts knowledge distillation to transfer knowledge from a full-sized "teacher-net" to the "student-net". However, it still suffers from high training cost.

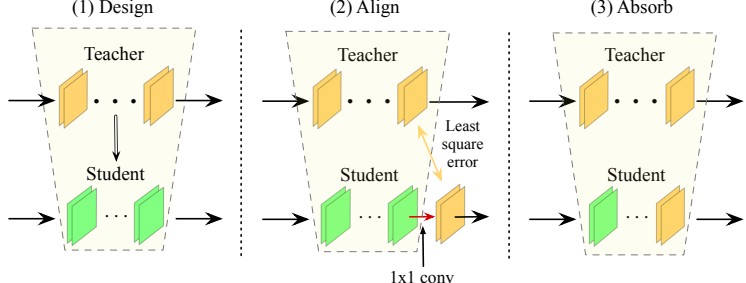

Figure 1: Three steps of our few-sample knowledge distillation. (1) design student-net from scratch or by compressing teacher-net; (2) add 1×1 conv-layer at the end of each block of student-net (before ReLU), and align teacher and student by estimating the parameter using least-squared regression; (3) absorb or merge the added 1×1 conv-layer into the previous conv-layer to obtain final student-net.

As is known, children can learn knowledge concept from adults with few examples. This cognition phenomenon has motivated the development of the few-shot learning (Fei-Fei et al., 2006; Bart & Ullman, 2005), which aims to learn information about object categories from a few training samples, and focuses more on image classification task. Nevertheless, it inspires people to consider the possibility of knowledge distillation from few samples. Some recent works on knowledge distillation address this problem by constructing "pseudo" training data (Kimura et al., 2018; Lopes et al., 2017) with complicated heuristics and heavy engineering, which are still costly.

This paper proposes a novel and simple three-step method for few-sample knowledge distillation (FSKD) as illustrated in Figure 1, including student-net design, teacher-student alignment, and absorbing added conv-layer. We assume that both "teacher" and "student" nets have the same feature map sizes at each corresponding block. However, the relatively small student-net can be obtained in various ways, such as pruning/decomposing the teacher-net, and fully redesigned network with random initialization. We add a 1×1 conv-layer at the end of each block of the student-net and align the block-level outputs between "teacher" and "student", which is done by estimating the parameters of the added layer with few samples using least square regression. Since the added 1×1 conv-layers have relatively few parameters, we can get a good approximation from a small number of samples. We further prove that the added 1×1 conv-layer can be absorbed/merged into the previous conv-layer when certain conditions fulfill, so that the new conv-layer has the same number of parameters and computation cost as the older/previous one.

We argue that FSKD has many potential applications, especially when fine-tuning or full training is not feasible in practice. We just name a few such cases below. *First*, edge devices have limited computing resources, while FSKD offers the possibility of on-device learning to compress deep models with a limited number of samples. *Second*, FSKD may help software/hardware vendors optimizing the deep models from their customers when full training data is unavailable due to privacy or confidential issues. *Third*, FSKD enables fast model deployment optimization when there is a strict time budget. Our major contributions can be summarized as follows:

(1) To the best of our knowledge, we are the first to show that knowledge distillation can be done with few samples within minutes on a desktop PC.

(2) The proposed FSKD method is widely applicable not only for fully redesigned student-nets but also for compressed networks from pruning and decomposition-based methods.

(3) We demonstrate significant performance improvement of the student-net by FSKD, comparing to existing distillation techniques on various datasets and network structures.

## 2 RELATED WORK

**Knowledge Distillation** (KD) transfers knowledge from a pre-trained large "teacher" network (or even an ensemble of networks) to a small "student" network, for facilitating the deployment at test time. Originally, this is done by regressing the softmax output of the teacher model (Hinton et al., 2015). The soft continuous regression loss used here provides richer information than the label based loss, so that the distilled model can be more accurate than training on labeled data with cross-entropy loss. Later, various works have extended this approach by matching other statistics, including intermediate feature responses (Romero et al., 2015; Chen et al., 2016), gradient (Srinivas & Fleuret, 2018), distribution (Huang & Wang, 2017), Gram matrix (Yim et al., 2017), etc. Deep mutual learning (Zhang et al., 2018) trains a cohort of student-nets and teaches each other collaboratively with model distillation throughout the training process. All these methods require a large amount of

data (known as the "transfer set") to transfer the knowledge, whereas we aim to provide a solution with a limited number of samples. We need *emphasize* that FSKD has a quite different philosophy on aligning intermediate responses to the closest knowledge distillation method FitNet (Romero et al., 2015). FitNet re-trains the whole student-net with intermediate supervision using a larger amount of data, while FSKD only estimates parameters for the added $1 \times 1$ conv-layer with few samples. We will verify in experiments that FSKD is not only more efficient but also more accurate than FitNet.

**Network Pruning** methods obtain a small network by pruning weights from a trained larger network, which can keep the accuracy of the larger model if the prune ratio is set properly. Han et al. (2015) proposes to prune the individual weights that are near zero. Recently, channel pruning has become increasingly popular thanks to its better compatibility with off-the-shelf computing libraries, compared with weights pruning. Different criteria have been proposed to select the channel to be pruned, including norm of weights (Li et al., 2016), scales of multiplicative coefficients (Liu et al., 2017), statistics of next layer (Luo et al., 2017), etc. It is usually required iterative loop between pruning and fine-tuning for achieving better pruning ratio and speedup. Similar gradually adjusting trick is also applied to train very-deep neural networks (Smith et al., 2016). Meanwhile, **Network Decomposition** methods try to factorize heavy layers in DNNs into multiple lightweight ones. For instance, it may adopt low-rank decomposition to fully-connection layers (Denton et al., 2014), and different kinds of tensor decomposition to conv-layers (Zhang et al., 2016; Kim et al., 2016). However, aggressive network pruning or network decomposition usually lead to large accuracy drops, thus fine-tuning is required to alleviate those drops (Li et al., 2016; Liu et al., 2017; Zhang et al., 2016). As aforementioned, KD is more accurate than directly training on labeled data, it is of great interest to explore KD on extremely pruned or decomposed networks, especially under the few-sample setting.

**Learning with few samples** has been extensively studied under the concept of one-shot or few-shot learning. One category of methods directly model few-shot samples with generative models (Fei-Fei et al., 2006; Lake et al., 2011), while most others study the problem under the notion of transfer learning (Bart & Ullman, 2005; Ravi & Larochelle, 2017). In the latter category, meta-learning methods (Vinyals et al., 2016; Finn et al., 2017) solve the problem in a learning to learn fashion, which has been recently gaining momentum due to their application versatility. Most studies are devoted to the image classification task, while it is still less-explored for knowledge distillation from few samples. Recently, some works tried to address this problem. Kimura et al. (2018) constructs pseudo-examples using the inducing point method, and develops a complicated algorithm to optimize the model and pseudo-examples alternatively. Lopes et al. (2017) records per-layer meta-data for the teacher-net in order to reconstruct a training set, and then adopts a standard training procedure to obtain the student-net. Both are very costly due to the complicated and heavy training procedure. On the contrary, we aim for a simple solution for knowledge distillation from few samples.

# 3 FEW-SAMPLE KNOWLEDGE DISTILLATION (FSKD)

## 3.1 OVERVIEW

Our FSKD method consists of three steps as shown in Figure 1. *First*, we design a student-net either by pruning/decomposing the teacher-net, or by fully redesigning a small student-net with random initialization. *Second*, we add a $1 \times 1$ conv-layer at the end of each block of the student-net and align the block-level outputs between "teacher" and "student" by estimating the parameters for the added layer from few samples. *Third*, we absorb the added $1 \times 1$ conv-layer into the previous conv-layer without introducing extra parameters and computations into the student-net.

Two reasons make this idea work efficiently. *First*, the $1 \times 1$ conv-layers have relatively few parameters, which do not require too many data for the estimation. *Second*, the block-level output from teacher-net provides rich information as shown in FitNet (Romero et al., 2015). Below, we will first provide the theoretical derivation why the added $1 \times 1$ conv-layer could be absorbed/merged into the previous conv-layer. Then we provide details on how we do the block-level output alignment.

## 3.2 ABSORBABLE $1 \times 1$ CONV-LAYER

Let's first give some mathematic notions for different kinds of convolutions before moving to the theoretical derivation. A *regular convolution* consists of multi-channel and multi-kernel filters which build both cross-channel correlations and spatial correlations. Formally, a regular convolution layer

can be represented by a 4-dimensional tensor $\mathbf{W} \in \mathbb{R}^{n_o \times n_i \times k \times k}$, where $n_o$ and $n_i$ are the number of output and input channels respectively, and $k \times k$ is the squared spatial kernel size. The *point-wise (PW) convolution*, also known as $1 \times 1$ convolution (Lin et al., 2014) can be represented by a tensor $\mathbf{P} \in \mathbb{R}^{n_o \times n_i \times 1 \times 1}$, which is actually degraded from a 4-dimensional tensor to a 2-dimensional matrix. The *depth-wise (DW) convolution* (Chollet, 2016) does per-channel 2D convolution for each input channel, so that it can be represented by a tensor $\mathbf{D} \in \mathbb{R}^{1 \times n_i \times k \times k}$. Due to no-correlation among output channels, it usually follows by a point-wise convolution to model their correlations. This combination (DW + PW) is also named as *depth-wise separable convolution* by Chollet (2016).

**Theorem 1.** *A pointwise convolution with tensor* $\mathbf{Q} \in \mathbb{R}^{n_o' \times n_i' \times 1 \times 1}$ *can be absorbed into the previous convolution layer with tensor* $\mathbf{W} \in \mathbb{R}^{n_o \times n_i \times 1 \times 1}$ *to obtain the **absorbed tensor** $\mathbf{W}' = \mathbf{Q} \circ \mathbf{W}$, where* $\circ$ *is absorbing operator and* $\mathbf{W}' \in \mathbb{R}^{n_o' \times n_i \times k \times k}$ *if the following conditions are satisfied.*

    *c1. The output channel number of* $\mathbf{W}$ *equals to the input channel number of* $\mathbf{Q}$, *i.e.,* $n_o = n_i'$.

    *c2. No non-linear activation layer like ReLU (Nair & Hinton, 2010) between* $\mathbf{W}$ *and* $\mathbf{Q}$.

Due to the space limitation, we put the proof and the detailed form of the absorbing operator in Appendix-A. The number of output channels of $\mathbf{W}'$ is $n_o'$, which is different from that of $\mathbf{W}$ (i.e., $n_o$). It is easy to have the following corollary.

**Corollary 1.** *When the following condition is satisfied for* $\mathbf{Q}$,

    *c3. the number of input and output channels of* $\mathbf{Q}$ *equals to the number of output channel of* $\mathbf{W}$, *i.e.,* $n_i' = n_o' = n_o$, $\mathbf{Q} \in \mathbb{R}^{n_o \times n_o \times 1 \times 1}$,

*the absorbed convolution tensor* $\mathbf{W}'$ *has the same parameters and computation cost as* $\mathbf{W}$, *i.e. both* $\mathbf{W}', \mathbf{W} \in \mathbb{R}^{n_o \times n_i \times k \times k}$.

This condition is required not only for ensuring the same parameter size and computing cost, but also for ensuring current layer output size matching/connectable to next layer input size.

## 3.3 BLOCK-LEVEL ALIGNMENT AND ABSORBING

Now we consider the knowledge distillation problem. Suppose $\boldsymbol{X}^s, \boldsymbol{X}^t \in \mathbb{R}^{n_o \times d}$ are the block-level output in matrix form for the student-net and teacher-net respectively, where $d$ is the per-channel feature map resolution size. We add a $1 \times 1$ conv-layer $\mathbf{Q}$ at the end of each block of student-net before non-linear activation, which satisfies condition $c1 \sim c3$. As $\mathbf{Q}$ is degraded to the matrix form, it can be estimated with least squared regression as

$$\mathbf{Q}^* = \arg\min_{\mathbf{Q}} \sum_{i=1}^N \|\mathbf{Q} * \mathbf{X}_i^s - \mathbf{X}_i^t\|, \tag{1}$$

where $N$ is the number of samples used, and "*" here means matrix product. The number of parameters of $\mathbf{Q}$ is $n_o \times n_o$, where $n_o$ is the number of output channels in the block, which is usually not too large so that we can estimate $\mathbf{Q}$ with a limited number of samples.

Suppose there are $M$ corresponding blocks in the teacher and student networks required to align, to achieve our goal, we need minimize the following loss function

$$\mathcal{L}(\mathbf{Q}_j) = \sum_{j=1}^M \sum_{i=1}^N \|\mathbf{Q}_j * \boldsymbol{X}_{ij}^s - \boldsymbol{X}_{ij}^t\|_F, \tag{2}$$

where $\mathbf{Q}_j$ is the tensor for the added $1 \times 1$ conv-layer of the $j$-th block. In practice, we optimize this loss with a block-coordinate descent (BCD) algorithm (Hong et al., 2017), which greedy handles each of the $M$ terms/blocks in Equation 2 in the student-net sequentially as shown in algorithm 1 at Appendix-B, instead of optimizing this loss all together using standard SGD. The BCD algorithm for FSKD has the following advantages:

    (1) The BCD algorithm processes each block greedy with a sequential update rule, and each $\mathbf{Q}$ can be solved much cheaper with a small number of samples by aligning the block-level responses between teacher and student networks, while SGD considers $\{\mathbf{Q}_j\}$ all together which theoretically requires more data.

    (2) The alignment procedure is very efficient, which can be usually done within several minutes for the entire network.

| | Top1-before(%) | Top1-after(%) | FLOPs($\times 10^8$) | Reduced | #Param($\times 10^6$) | Pruned | #Samples |
|---|---|---|---|---|---|---|---|
| VGG-16 | 92.66 | - | 3.11 | - | 15 | - | - |
| Scheme-A + FSKD-BCD | 85.42 | 92.37 | 2.06 | 34% | 5.3 | 64% | 100 |
| Scheme-A + FSKD-SGD | 85.42 | 92.18 | 2.06 | 34% | 5.3 | 64% | 100 |
| Scheme-A + FitNet | 85.42 | 91.23 | 2.06 | 34% | 5.3 | 64% | 100 |
| Scheme-A + FSKD-BCD | 85.42 | 92.46 | 2.06 | 34% | 5.3 | 64% | 500 |
| Scheme-A + FSKD-SGD | 85.42 | 92.42 | 2.06 | 34% | 5.3 | 64% | 500 |
| Scheme-A + FitNet | 85.42 | 92.13 | 2.06 | 34% | 5.3 | 64% | 500 |
| Scheme-A + Fine-tuning | 85.42 | 90.25 | 2.06 | 34% | 5.3 | 64% | 500 |
| Scheme-A + Full fine-tuning | 85.42 | 92.54 | 2.06 | 34% | 5.3 | 64% | 50000 |
| Scheme-B + FSKD-BCD | 47.90 | 90.17 | 1.33 | 58% | 3.4 | 77% | 100 |
| Scheme-B + FSKD-SGD | 47.90 | 89.41 | 1.33 | 58% | 3.4 | 77% | 100 |
| Scheme-B + FitNet | 47.90 | 88.76 | 1.33 | 58% | 3.4 | 77% | 100 |
| Scheme-B + FSKD-BCD | 47.90 | 91.21 | 1.33 | 58% | 3.4 | 77% | 500 |
| Scheme-B + FSKD-SGD | 47.90 | 90.76 | 1.33 | 58% | 3.4 | 77% | 500 |
| Scheme-B + FitNet | 47.90 | 90.68 | 1.33 | 58% | 3.4 | 77% | 500 |
| Scheme-B + Fine-tuning | 47.90 | 83.36 | 1.33 | 58% | 3.4 | 77% | 500 |
| Scheme-B + Full fine-tuning | 47.90 | 91.53 | 1.33 | 58% | 3.4 | 77% | 50000 |

Table 1: Performance comparison between FitNet, fine-tuning (Li et al., 2016), FSKD with student-nets from **filter pruning** of VGG-16 with scheme-A/B on CIFAR-10. "Full fine-tuning" uses full training data.

(3) The alignment procedure itself does not require class label information of input data due to its regression nature. However, if we fully redesign the student-net from scratch with random weights, we may leverage SGD on a few labeled samples to initialize the network. Our FSKD can still produce significant performance gains over SGD in this case.

(4) Our FSKD works extremely well for student-net obtained by aggressively pruning/decomposing the teacher-net. It beats the standard fine-tuning based solution on the number of data required, processing speed, and accuracy of the output student-net.

# 4 EXPERIMENT

We perform extensive experiments on different image classification datasets to verify the effectiveness of FSKD on various student-net construction methods. Student-nets can be obtained either from compressing the teacher-net or redesigning network structure with random initialization (termed "zero student network"). For the former case, we evaluate FSKD on three well-known compression methods, filter pruning (Li et al., 2016), network slimming (Liu et al., 2017), and network decoupling (Guo et al., 2018). We implement the code with PyTorch, and conduct experiments on a desktop PC with Intel i7-7700K CPU and one NVidia 1080TI GPU.

## 4.1 STUDENT NETWORK FROM COMPRESSING TEACHER NETWORK

### FILTER PRUNING

We first obtain the student-nets using the filter pruning method (Li et al., 2016), which prunes out conv-filters according to the $L_1$ norm of their weights. The $L_1$ norm of filter weights are sorted and the smallest portion of filters will be pruned to reduce the number of filter-channels in a conv-layer.

We make a comprehensive study of VGG-16 (Simonyan & Zisserman, 2015) on CIFAR-10 dataset to evaluate the performance of FSKD along with different configuration settings. Following Li et al. (2016), we first prune half of the filters in conv1_1, conv4_1, conv4_2, conv4_3, conv5_1, conv5_2, conv5_3 while keeps the other layer unchanged (*scheme-A*). We also propose another more aggressive pruning scheme named *scheme-B*, which pruned 10% more filters in the aforementioned layers, and also pruned 20% filters for the remaining layers. Scheme-A prunes 64% of total parameters with 7% accuracy drop. Scheme-B prunes 77% of total parameters with almost 50% accuracy drop. We use those two pruned networks as student-nets in this study. [1]

We illustrate in Figure 6 in Appendix-C how teacher and student are aligned at block-level. For the few-sample setting, we randomly select 100 (10 for each category) and 500 (50 for each category) images from the CIFAR-10 training set, and keep them fixed in all experiments. Table 1 lists the results of different methods of recovering a pruned network, including FitNet (Romero et al., 2015), fine-tuning with limited data and full training data (Li et al., 2016). Note that we optimize FSKD with two algorithms: FSKD-BCD uses the BCD algorithm on block-level and FSKD-SGD optimizes the loss (Equation 2) all together with SGD algorithm. In the BCD algorithm, we do not observe benefit for the iteration number $T > 1$ over $T = 1$, so that we set $T = 1$ in all our following experiments. This is consistent with the finding by (Hong et al., 2017) that the convergence is sublinear when each

---

[1]An extremely pruned case "scheme-C" is also provided in Appendix-D.

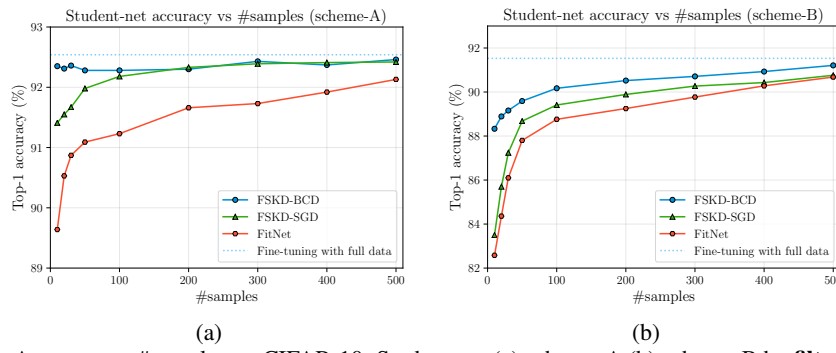

Figure 2: Accuracy vs #samples on CIFAR-10. Student-net (a) scheme-A (b) scheme-B by **filter pruning**.

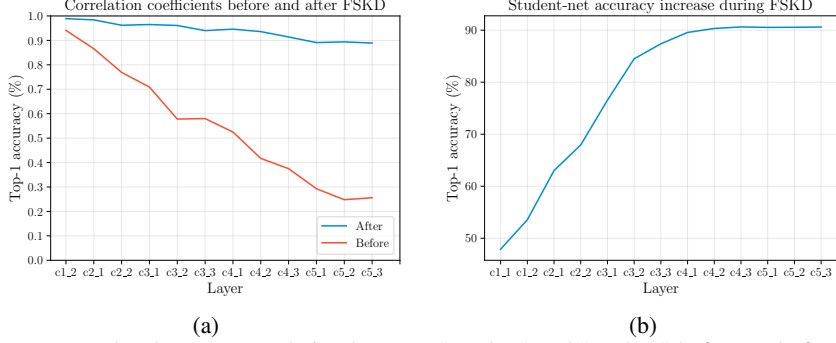

Figure 3: (a) Layer-level output correlation between "teacher" and "student" before and after FSKD on student-nets (scheme-A) by **filter pruning**. (b) Accuracy change during sequentially block-level alignment.

block is minimized exactly (here due to linear structure). Regarding the processing speed, FSKD-BCD can be done in **19.3** seconds for student-net from scheme-B with 500 samples, while FitNet requires **157.3** seconds when converged, which is about $8.1\times$ slower. This verifies our previous claim that FSKD is more efficient than FitNet. It can be seen that in the few-sample setting, FSKD-BCD provides better accuracy recovery than both FitNet and the fine-tuning procedure adopted in Li et al. (2016). For instance, for scheme-B with only 500 samples, FSKD can recover the accuracy from 47.9% to 91.2%, while few-sample fine-tuning can only recover the accuracy to 83.36%. When full training set available, it will take about 30 minutes for full fine-tuning to reach similar accuracy as FSKD. This demonstrates the big advantages of FSKD over full fine-tuning based solutions.

Figure 2 further studies the performance with different amount of training samples available. It can be observed that our FSKD-BCD keep outperforming FSKD-SGD, FitNet under the same training samples. In particular, FSKD-SGD and FitNet experience a noticeable accuracy drop when the number of samples is less than 100, while FSKD-BCD can still recover the accuracy of the pruned network to a high level. It is also interesting to note that fine-tuning experiences even larger accuracy drops than FitNet when the data amount is limited. This verifies that knowledge distillation methods like FitNet provide richer information than fine-tuning/re-training with label based loss.

As is shown, FSKD-BCD performs better and tends to be more sample-efficient than FSKD-SGD. Therefore, we choose it as the default algorithm, and denote it as FSKD for simplification in the following studies. We further illustrate the per-layer (block) feature responses difference between teacher and student before and after using FSKD in Figure 3a. Before applying FSKD, the correlation between teacher and student is broken due to the aggressive compression. However, after FSKD, the per-layer correlations between teacher and student are restored. This verifies the ability of FSKD for recovering lost information. We do see a decreasing trend with layer depth increased, which is possibly due to error accumulation through multiple convolutional layers. We also show the accuracy change during sequentially block-level alignment in Figure 3b, which clearly demonstrate the effectiveness of our sequentially block-by-block update in the BCD algorithm.

### NETWORK SLIMMING

We then study the student-net from another channel pruning method named network slimming (Liu et al., 2017), which removes insignificant filter channels and corresponding feature maps using sparsified channel scaling factors. Network slimming consists of three steps: sparse regularized training, pruning and fine-tuning. Here, we replace the time-consuming fine-tuning step with our

| Channel-prune-ratio | Top1-before(%) | Top1-after(%) | FLOPs($\times 10^8$) | Reduced | #Param($\times 10^6$) | Pruned |
|---|---|---|---|---|---|---|
| VGG-19 | 93.38 | - | 7.97 | - | 20 | - |
| 70% + FSKD | 15.90 | 93.41 | 3.91 | 51% | 2.2 | 89% |
| 70% + FitNet | 15.90 | 90.47 | 3.91 | 51% | 2.2 | 89% |
| 70% + Fine-tuning | 15.90 | 62.86 | 3.91 | 51% | 2.2 | 89% |
| ResNet-164 | 95.07 | - | 4.99 | - | 1.7 | - |
| 60% + FSKD | 54.46 | 94.19 | 2.75 | 45% | 1.1 | 37% |
| 60% + FitNet | 54.46 | 88.94 | 2.75 | 45% | 1.1 | 37% |
| 60% + Fine-tuning | 54.46 | 60.94 | 2.75 | 45% | 1.1 | 37% |
| DenseNet-40 | 94.18 | - | 5.33 | - | 1.1 | - |
| 60% + FSKD | 88.24 | 93.62 | 2.89 | 46% | 0.5 | 54% |
| 60% + FitNet | 88.24 | 91.37 | 2.89 | 46% | 0.5 | 54% |
| 60% + Fine-tuning | 88.24 | 88.98 | 2.89 | 46% | 0.5 | 54% |

Table 2: Performance comparison between FSKD, FitNet and fine-tuning (Liu et al., 2017) on different network structures obtained by **network slimming** with 100 samples randomly selected from CIFAR-10 training set.

| Channel-prune-ratio | Top1-before(%) | Top1-after(%) | FLOPs($\times 10^8$) | Reduced | #Param($\times 10^6$) | Pruned |
|---|---|---|---|---|---|---|
| VGG-19 | 72.08 | - | 7.97 | - | 20 | - |
| 50% + FSKD | 9.24 | 71.98 | 5.01 | 37% | 5.0 | 75% |
| 50% + FitNet | 9.24 | 69.52 | 5.01 | 37% | 5.0 | 75% |
| 50% + Fine-tuning | 9.24 | 48.75 | 5.01 | 37% | 5.0 | 75% |
| ResNet-164 | 76.56 | - | 5.00 | - | 1.7 | - |
| 40% + FSKD | 46.07 | 76.11 | 3.33 | 33% | 1.5 | 14% |
| 40% + FitNet | 46.07 | 73.87 | 3.33 | 33% | 1.5 | 14% |
| 40% + Fine-tuning | 46.07 | 57.45 | 3.33 | 33% | 1.5 | 14% |
| DenseNet-40 | 73.21 | - | 5.33 | - | 1.1 | - |
| 40% + FSKD | 60.62 | 73.26 | 3.71 | 30% | 0.71 | 36% |
| 40% + FitNet | 60.62 | 71.08 | 3.71 | 30% | 0.71 | 36% |
| 40% + Fine-tuning | 60.62 | 62.36 | 3.71 | 30% | 0.71 | 36% |

Table 3: Performance comparison between FSKD, FitNet and fine-tuning (Liu et al., 2017) on different network structures obtained by **network slimming** with 500 samples randomly selected from CIFAR-100 training set.

FSKD, and follow the original paper (Liu et al., 2017) to conduct experiments to prune different networks on different datasets. We apply FSKD on networks pruned from VGG-19, ResNet-164, and DenseNet-40 (Huang et al., 2017), on both CIFAR-10 and CIFAR-100 datasets. Table 2 lists results on CIFAR-10, while Table 3 lists results on CIFAR-100. Note that the channel-prune-ratio (like 70% in Table 2) means the portion of channels that are removed in comparison to the total number of channels in the network. It shows that FSKD consistently outperforms FitNet and fine-tuning with a notable margin under the few-sample setting on all evaluated networks and datasets.

### NETWORK DECOUPLING

Network decoupling (Guo et al., 2018) decomposes a regular convolution layer into the sum of several blocks, where each block consists of a depth-wise (DW) convolution layer and a point-wise (PW, $1\times1$) convolution layer. The ratio of compression increases as the number of blocks decreases, but the accuracy of the compressed model will also drop. Since each decoupled block ends with a $1\times1$ convolution, we can apply FSKD at the end of each decoupled block.

Following (Guo et al., 2018), we obtain student-nets by decoupling VGG-16 and ResNet-18 pre-trained on ImageNet with different $T$ values, where $T$ stands for the number of DW + PW blocks that a conv-layer decouples out. Figure 7 in Appendix-C illustrates how teacher and student are aligned at block-level in this case. For VGG-16, we also decouple half of the conv-layer with $T = 1$ and the other half $T = 2$, and denote the case as "$T = 1$mix". We evaluate the resulted network performance on the validation set of the ImageNet classification task. We randomly select one image from each of the 1000 classes in ImageNet training set to obtain 1000 samples as our FSKD training set. Table 4 shows the top-1 accuracy of student-net before and after applying FSKD on VGG-16 and ResNet-18.

It is quite interesting to see that in the case of $T = 1$mix for VGG-16 and $T = 2$ for ResNet-18, we can recover the accuracy of student-net from nearly random guess (0.12%, 0.21%) to a much higher level (51.3% and 49.5%) with only 1000 samples. In all the other cases, FSKD can recover the accuracy of a highly-compressed network to be comparable with the original network. One possible explanation is that the highly-compressed networks still inherit some representation power from the teacher-net i.e., the depth-wise $3\times3$ convolution, while lacking the ability to output meaningful predictions due to the inaccurate/degraded $1 \times 1$ convolution. The FSKD calibrates the $1 \times 1$ convolution by aligning the block-level responses between "teacher" and "student" so that the lost information in $1 \times 1$ convolution is compensated, and reasonable recovery is achieved [2].

---

[2]We thus make a bold hypothesis that point-wise is more critical for performance than depthwise, so that even depthwise $3 \times 3$ conv-layers are initialized to be orthogonal from random data, training only pointwise conv-layers could provide enough accurate results. We verify this in Appendix-E.

|  | Top1-before(%) | Top1-after (%) | GFLOPs | Reduced | #Param*($\times 10^6$) | Pruned |
|---|---|---|---|---|---|---|
| VGG-16 (teacher) | 68.4 | - | 15.47 | - | 14.71 | - |
| Decoupled ($T = 2$) + FSKD | 0.24 | 62.7 | 3.76 | 75.7% | 3.35 | 77.2% |
| Decoupled ($T = 3$) + FSKD | 1.57 | 67.1 | 5.54 | 64.2% | 5.02 | 65.8% |
| Decoupled ($T = 4$) + FSKD | 54.6 | 67.6 | 7.31 | 52.7% | 6.69 | 54.5% |
| ResNet-18 (teacher) | 67.1 | - | 1.83 | - | 11.17 | - |
| Decoupled ($T = 2$) + FSKD | 0.21 | 49.5 | 0.55 | 70.0% | 2.69 | 75.9% |
| Decoupled ($T = 3$) + FSKD | 3.99 | 61.9 | 0.75 | 59.0% | 3.95 | 64.6% |
| Decoupled ($T = 4$) + FSKD | 26.5 | 65.1 | 0.95 | 48.1% | 5.20 | 53.4% |
| Decoupled ($T = 5$) + FSKD | 53.6 | 66.3 | 1.15 | 37.2% | 6.46 | 42.2% |

Table 4: Performance of FSKD on different student nets obtained by **network decoupling** VGG-16 and ResNet-18 with different parameters $T$ on ImageNet dataset. "$*$" here means that parameters from FC-layer are not counted, only those from conv-layers are counted, since decoupling only handles the conv-layers.

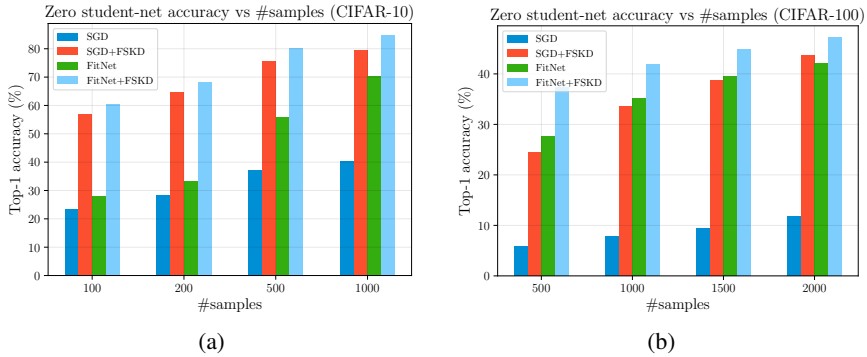

(a)                           (b)

Figure 4: Accuracy vs #samples for **zero student network** on (a) CIFAR-10, (b) CIFAR-100.

## 4.2 ZERO STUDENT NETWORK

Finally, we evaluate FSKD on fully redesigned student-net with a different structure from the teacher and random initialized parameters (named as zero student-net). We conduct experiments on CIFAR-10 and CIFAR-100 with VGG-19 as the teacher-net and a shallower VGG-13 as the student-net. Due to the similar structure between VGG-13 and VGG-19, they can be easily aligned in block-level.

The random initialized network does not contain any information about the training set. Simply training this network using SGD with few samples will lead to poor generalization ability, as shown in Figure 4. We propose two schemes to combine FSKD in the training procedure: *SGD+FSKD* and *FitNet+FSKD*. In the SGD+FSKD case, we first use SGD to train the student-net (without using teacher-net information) on the given few labeled samples with 150 epochs (multi-step learning-rate decay at every 50 epochs from 0.01 to 0.0001), and then apply FSKD to the obtained student-net using the same few-sample set. We repeat these two steps until the training loss converges. In the FitNet+FSKD case, we keep the same few-sample set, and simply replace the SGD with FitNet (using teacher-net information) to add supervision on intermediate responses during training.

We compare the results from four different recovery methods: running SGD until convergence, SGD+FSKD, running FitNet until convergence, and FitNet+FSKD. In order to better simulate the few-sample setting, we do not apply data augmentation to the training set. We randomly pick 100, 200, 500, 1000 samples from the CIFAR-10 training set, and 500, 1000, 1500, 2000 samples from the CIFAR-100 training set, and keep these few-sample sets fixed in this study. Figure 4 shows the comparison results on the four methods and four few-sample sets. It shows that FSKD+SGD takes a big jump over pure SGD, and FSKD+FitNet also takes a big jump over pure FitNet. FSKD+SGD performs much better than FitNet on CIFAR-10, while this is not true on CIFAR-100. There are two possible reasons. *First*, we did not enable data augmentation so that few-sample SGD is underfitting, which provides much less information than what the student-net can get from the teacher-net in FitNet. *Second*, CIFAR-100 is much more difficult than CIFAR-10 so that the performance is more sensitive to the number of samples. However, FSKD+SGD can still achieve accuracy on par with FitNet. We should also note here that the zero-student-nets have accuracy gaps with the fully-trained teacher-nets on both CIFAR-10 and CIFAR-100. This is reasonable and acceptable, considering that we did not use data augmentation and trained the model with very few samples. Nevertheless, it still demonstrates the advantages of our FSKD over SGD and FitNet based methods. To further illustrate the benefit of FSKD over SGD, we visualize the convolution kernel (in terms of decoupled pointwise convolutions) before SGD, after SGD, and after SGD+FSKD in Appendix-F.

## 5 CONCLUSION

We proposed a novel and simple method for knowledge distillation from few samples (FSKD). The method works for student-nets constructed in various ways, including compression from teacher-nets and fully redesigned networks with random initialization on various datasets. Experiments demonstrate that FSKD outperforms existing knowledge distillation methods by a large margin in the few-sample setting, while requires significantly less computation budget. This advantage will bring many potential applications and extensions for FSKD.

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

APPENDIX-A: PROOF OF THEOREM 1

*Proof.* When $\mathbf{W}$ is a point-wise convolution with tensor $\mathbf{W} \in \mathbb{R}^{n_o \times n_i \times 1 \times 1}$, both $\mathbf{W}$ and $\mathbf{Q}$ are degraded into matrix form. It is obvious that when condition $c1 \sim c3$ satisfied, the theorem holds with $\mathbf{W}' = \mathbf{Q} * \mathbf{W}$ in this case, where $*$ indicates matrix multiplication.

When $\mathbf{W}$ is a regular convolution with tensor $\mathbf{W} \in \mathbb{R}^{n_o \times n_i \times k \times k}$, the proof is non-trivial. Fortunately, recent work on network decoupling (Guo et al., 2018) presents an important theoretic result as the basis of our derivation.

**Lemma 1.** *Regular convolution can be exactly expanded to a sum of several depth-wise separable convolutions. Formally,* $\forall\, \mathbf{W} \in \mathbb{R}^{n_o \times n_i \times k \times k}$, $\exists\, \{\mathbf{P}_k, \mathbf{D}_k\}_{k=1}^K$, *where* $\mathbf{P}_k \in \mathbb{R}^{n_o \times n_i \times 1 \times 1}$, $\mathbf{D}_k \in \mathbb{R}^{1 \times n_i \times k \times k}$,

$$s.t. \ (a) K \leq k^2; \tag{3}$$
$$(b) \mathbf{W} = \sum\nolimits_{k=1}^K \mathbf{P}_k \circ \mathbf{D}_k,$$

*where $\circ$ is the compound operation, which means performing $\mathbf{D}_k$ before $\mathbf{P}_k$.*

Please refer to Guo et al. (2018) for the details of proof for this Lemma. When $\mathbf{W}$ is applied to an input patch $\mathbf{x} \in \mathbb{R}^{n_i \times k \times k}$, we obtain a response vector $\mathbf{y} \in \mathbb{R}^{n_o}$ as

$$\mathbf{y} = \mathbf{W} \otimes \mathbf{x}, \tag{4}$$

where $y_o = \sum_{i=1}^{n_i} W_{o,i} \otimes x_i, o \in [n_o], i \in [n_i]$, and $\otimes$ here means convolution operation. $W_{o,i} = \mathbf{W}[o,i,:,:]$ is a tensor slice along the $i$-th input and $o$-th output channels, $x_i = \mathbf{x}[i,:,:]$ is a tensor slice along the $i$-th channel of 3D tensor $\mathbf{x}$.

When point-wise convolution $\mathbf{Q}$ is added after $\mathbf{Q}$ without non-linear activation between them, we have

$$\mathbf{y}' = \mathbf{Q} \circ (\mathbf{W} \otimes \mathbf{x}). \tag{5}$$

With Lemma-1, we have

$$\mathbf{y}' = (\mathbf{Q} \circ \sum\nolimits_{k=1}^K \mathbf{P}_k \circ \mathbf{D}_k) \otimes \mathbf{x} = (\sum\nolimits_{k=1}^K (\mathbf{Q} * \mathbf{P}_k) \circ \mathbf{D}_k) \otimes \mathbf{x} \tag{6}$$

As both $\mathbf{Q}$ and $\mathbf{P}_k$ are degraded into matrix form, denoting $\mathbf{P}'_k = \mathbf{Q} * \mathbf{P}_k$ and $\mathbf{W}' = \sum_{k=1}^K \mathbf{P}'_k \circ \mathbf{D}_k$, we have $\mathbf{y}' = \mathbf{W}' \circ \mathbf{x}$. This proves the case when $\mathbf{W}$ is a regular convolution. □

APPENDIX-B: ALGORITHM OF BLOCK-LEVEL ALIGNMENT FOR FSKD

The block-level alignment algorithm for FSKD is in fact a block-coordinate descent (BCD) algorithm with greedy sequential block-level update rule, as shown in algorithm 1.

---
**Algorithm 1:** Block-coordinate descent algorithm for FSKD
---
1 **Data:** Given student-net $s$ and teacher-net $t$ and input data $\{\boldsymbol{X}_i\}_{i=1}^N$,
2     number of aligned blocks $M$, number of iterations $T$
3 **for** $k = 1 : T$ **do**
4     Random flip input dataset to obtain $\{\boldsymbol{X}'_i\}_{i=1}^N$;
5     **for** $j = 1 : M$ **do**
6        Feed $\{\boldsymbol{X}'_i\}_{i=1}^N$ to the end of $j$-th block for teacher-net $t$ and and current student-net $s$;
7        Obtain response $\{\boldsymbol{X}^t_{ij}\}$ and $\{\boldsymbol{X}^s_{ij}\}$ from $j$-th block;
8        Add $1 \times 1$ conv-layer with tensor $\mathbf{Q}_j$ to the end of $j$-th block of student-net $s$ (before ReLU);
9        Solve $\mathbf{Q}_j$ with least-square regression based on Equation 1;
10        Merge $\mathbf{Q}_j$ into previous conv-layer $L_j$ with tensor $\mathbf{W}_j$ to obtain new tensor $\mathbf{W}'_j$ based on Theorem 1 for student-net;
11        Update $j$-th block of current student-net $s$;
12     **end**
13 **end**
14 **Result:** Absorbed conv-layers $\{\mathbf{W}'_j\}_{j=1}^M$ and updated student-net $s'$.
---

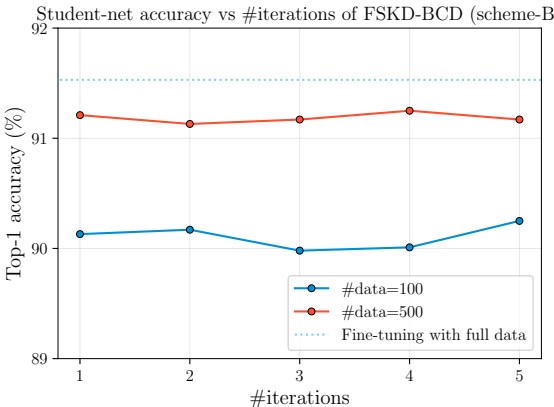

Figure 5: Accuracy vs #iterations of FSKD-BCD on CIFAR-10. Student-net is scheme-B by **filter pruning**.

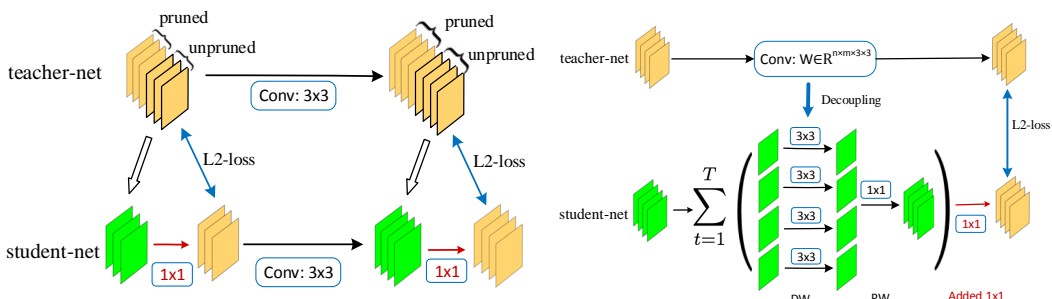

Figure 6: Illustration of FSKD on filter pruning and network slimming. At each block, we copy weights of the unpruned part in teacher-net to student-net, and align the feature maps of student-net to those unpruned feature maps of teacher-net by adding a $1 \times 1$ conv-layer (red-color) with L2-loss. The added $1 \times 1$ can be merged into the previous conv-layer in student-net.

Figure 7: Illustration of FSKD on network decoupling. At each block, we decouple regular-conv in teacher-net into a sum of depthwise + pointwise conv-layers as the block of student-net, and align the feature maps of student-net to that of teacher-net by adding a $1 \times 1$ conv-layer (red-color) with L2-loss. The added $1 \times 1$ can be merged into previous the pointwise layer in student-net.

The accuracy of FSKD-BCD versus the number of iterations $T$ is illustrated in Figure 5, which shows that more iterations do not bring noticeable performance gain. This is because in each iteration, the sub-problem is a linear optimization problem so that we can find exact minimization. This is consistent with the finding by (Hong et al., 2017). Therefore, in the paper, we only report the accuracy of $T = 1$ for FSKD-BCD.

## APPENDIX-C: ILLUSTRATION OF FSKD ON PRUNING AND DECOUPLING

Figure 6 illustrates how FSKD work for block-level alignment on the filter pruning (Li et al., 2016) and network slimming (Liu et al., 2017) cases. Figure 7 illustrates how FSKD works for block-level alignment on the network decoupling (Guo et al., 2018) case.

## APPENDIX-D: ITERATIVE PRUNING AND FSKD

Previous works show that one time extremely pruning may yield the pruned network unable to recovery from fine-tuning, while the iteratively pruning and fine-tuning procedure is observed effective to obtain extreme model compression (Han et al., 2016; Li et al., 2016; Liu et al., 2017). Inspired by these works, we proposed the iteratively pruning and FSKD procedure as described in algorithm 2 to achieve extremely compression rate. This solution is still much more efficient than iteratively pruning and fine-tuning due to the great efficiency of FSKD over fine-tuning.

---

**Algorithm 2:** Iteratively pruning and FSKD Algorithm

---

1  **Data:** Teacher-net $t$, input data $\{X_i\}_{i=1}^{N}$, prune-ratio-list $\{r_k\}_{k=1}^{K}$, number of iterations $T$
2  $s_{max} = \varnothing$;
3  **for** $t = 1 : T$ **do**
4     $q_{max} = 0$;
5     **for** $k = 1 : K$ **do**
6         Prune $s$ with ratio $r_k$ to obtain student-net $t$;
7         Run FSKD (algorithm 1) with $s$, $t$ and $\{X_i\}_{i=1}^{N}$, output $s'$;
8         Evaluation $s'$ on validation set to obtain score $q_k$;
9         **if** $q_k > q_{mqx}$ **then**
10             $q_{max} = q_k$;
11             $s_{max} = s'$;
12         **end**
13     **end**
14     Update teacher $t = s_{max}$;
15  **end**
16  **Result:** final student-net $s_{max}$.

---

| | Top1-before(%) | Top1-after(%) | FLOPs($\times 10^8$) | Reduced | #Param($\times 10^6$) | Pruned | #Samples |
|---|---|---|---|---|---|---|---|
| VGG-16 | 92.66 | - | 3.11 | - | 15 | - | - |
| Scheme-C + FSKD-BCD | 13.05 | 89.55 | 1.09 | 65% | 1.8 | 88% | 100 |
| Scheme-C + FSKD-SGD | 13.05 | 89.01 | 1.09 | 65% | 1.8 | 88% | 100 |
| Scheme-C + FitNet | 13.05 | 85.09 | 1.09 | 65% | 1.8 | 88% | 100 |
| Scheme-C + FSKD-BCD | 13.05 | 90.41 | 1.09 | 65% | 1.8 | 88% | 500 |
| Scheme-C + FSKD-SGD | 13.05 | 90.12 | 1.09 | 65% | 1.8 | 88% | 500 |
| Scheme-C + FitNet | 13.05 | 88.31 | 1.09 | 65% | 1.8 | 88% | 500 |
| Scheme-C + Fine-tuning | 13.05 | 78.13 | 1.09 | 65% | 1.8 | 88% | 500 |
| Scheme-C + Full fine-tuning | 13.05 | 90.77 | 1.09 | 65% | 1.8 | 88% | 50000 |

Table 5: Performance comparison between FitNet, fine-tuning (Li et al., 2016), FSKD-BCD/SGD with student-nets from **filter pruning** of VGG-16 with scheme-C on CIFAR-10. "Full fine-tuning"uses full training data.

Based on this procedure, we extremely prune VGG-16 on CIFAR-10 by 88% total parameters. Table 5 list the results comparison to fine-tuning, FitNet, etc. It verfies the effectiveness of our FSKD on this extremely pruned case.

## APPENDIX-E: TRAINING ONLY POINTWISE CONV-LAYER IS ACCURATE ENOUGH

People may challenge that learning $1 \times 1$-conv may loss representation power and ask why the added $1 \times 1$ convolution works so well with such few samples. According to the network decoupling theory (Lemma-1), any regular conv-layer could be decomposed into a sum of depthwise separable blocks, where each depthwise separable block consists of a depthwise (DW) convolution (for spatial correlation modeling) followed by a pointwise (PW) convolution (for cross-channel correlation modeling). The added $1 \times 1$ conv-layer is absorbed/merged into the previous PW layer finally. The decoupling yields that the number of parameters in PW-layer occupies most (>=80%) parameters of the whole network. We argue that learning $1 \times 1$-conv is still very powerful, and make a **bold hypothesis** in subsection 4.1 that PW conv-layer is more critical for performance than DW conv-layer. To verify this hypothesis, we make experiments on VGG16 and ResNet50 on CIFAR-10 and CIFAR-100 under below different settings.

(1) We train the network from random initialization with 150 epoches.
(2) We start from a random initialized network (VGG16 or ResNet50), and do full rank decoupling ($K = k^2$ in Equation 3) so that channels in DW layers are orthogonal, and PW layers are still fully random. Note that Lemma-1 ensures the network before and after decoupling are equivalent (i.e., able to transfer back and force from each other). We keep all the DW-layers fixed (with random orthogonal basis), and train only the PW layers with 150 epochs. We denote this scheme as ND-1*1.

Note that except the setting explicitly described, all the other configurations (including training epochs, hyper-parameters, hardware platform, etc) are kept the same on both experimental cases. Table 6 lists the experimental results on these two cases on both datasets with two different network structures. It is obvious that the 2nd case (ND-1*1) clearly outperforms the 1st case. This verifies our

| Model | CIFAR-10(%) | CIFAR-100(%) |
|---|---|---|
| VGG16 | 93.00 | 73.35 |
| VGG16 (ND-1*1) | 93.91 | 73.61 |
| ResNet50 | 92.64 | 69.93 |
| ResNet50 (ND-1*1) | 93.51 | 70.83 |

Table 6: Results by two schemes (1) full training (2) only training pointwise conv-layers (ND-1*1).

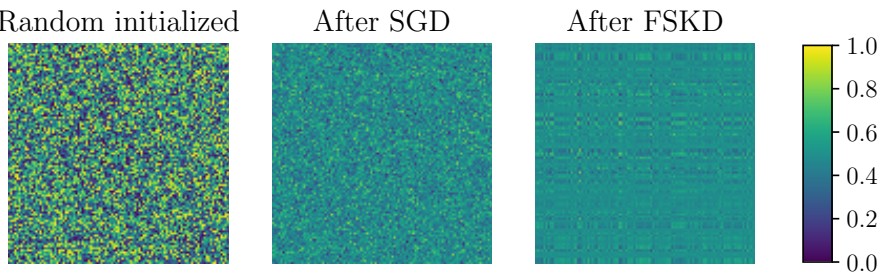

Figure 8: Decouple VGG13 into depthwise (DW) and pointwise (PW) conv-layers, and show one PW layer before SGD with random initialization (left), after SGD (middle), and after FSKD (right). Note values of the PW tensor are scaled into the range (0,1.0) by the min/max values of the tensor for better visulization.

hypothesis that when keeping DW channels orthogonal, training only the pointwise $(1 \times 1)$ conv-layer is accurate enough, or even better than training all the parameters together.

## APPENDIX-F: FILTER VISUALIZATION ON ZERO STUDENT-NET

To help better understanding how FSKD impacts the filters, we try to visualize the filter kernels. As the regular conv-layer kernel size is just $3 \times 3$ in the zero student-net (VGG13), it is hard to see a difference in such a small kernel-size. Instead, we consider visualizing the pointwise convolution tensor (degraded to matrix form) in Figure 8 for the following three cases:

(a) We initialize VGG13 with the MSRA initialization method, and then decouple one layer (64 input channels and 64 output channels) to obtain the PW conv-layer. For simplicity, we only visualize the PW tensor of the first decoupling block (in the left), which has size $64 \times 64$;

(b) We run SGD on few samples for VGG13 from random initialization until convergence, and then decouple the same layer to obtain the first-rank PW tensor (visualized in the middle);

(c) We further run FSKD on few samples for VGG13 already optimized by SGD, and then decouple the same layer to obtain the first-rank PW tensor (visualized in the right).

It shows that the tensor before SGD is fairly random on the value range, the tensor after SGD is less random, while the tensor after FSKD further shows some regular patterns, which indicates that there are some strong correlations among depthwise channels.

