# OpenReview forum: "Knowledge Distillation from Few Samples"
_ICLR.cc/2019/Conference_

### Official Review · AnonReviewer3 · 2018-10-28
**A practical method**

**Rating:** 6
**Confidence:** 3

**Review:**

In this paper, an efficient re-training algorithm for neural networks is proposed. The essence is like Hinton's distillation, but in addition to use the output of the last layer, the outputs of intermediate layers are also used. The core idea is to add 1x1 convolutions to the end of each layer and train them by fixing other parameters. Since the number of parameters to train is small, it performs well with the small number of samples such as 500 samples.

The proposed method named FKSD is simple yet achieves good performance. Also, it performs well with a few samples, which is desirable in terms of time complexity.

The downside of this paper is that there is no clear explanation of why the FKSD method goes well. For me, adding 1x1 convolution after the original convolution and fitting the kernel of the 1x1 conv instead of the original kernel looks a kind of reparametrization trick. Of course, learning 1x1 conv is easier than learning original conv because of a few parameters. However, it also restricts the representation power so we cannot say which one is always better. Do you have any hypothesis of why 1x1 conv works so well?



Minor:

The operator * in (1) is undefined.

What does the boldface in tables of the experiments mean? I was confused because, in Table 1, the accuracy achieved by FKSD is in bold but is not the highest one.

---

> ### Author Response · Authors · 2018-11-26
> **Response to Reviewer3**
>
> Thank you for your review and suggestions, we provide our response as follows.
>
> Q: Learning 1*1-conv restrict the representation power, any hypothesis why the added 1x1 works so well?
> As the added 1*1 conv-layer is finally absorbed into the decoupled 1*1 conv-layer, we here hypothesize that pointwise (1*1) convolution is more critical for performance than the depthwise convolution since it occupies >=80% parameters of the decoupled network. We design an experiment in Appendix-E to verify this hypothesis by comparing the full training to only training the 1*1 conv-layer with 3*3 initialized to be orthogonal from random data, on CIFAR-10/100 with VGG16 and ResNet50. The results show that full training works noticeably worse than our designed case. This interesting result verifies our hypothesis, and may inspire more researches to further understand CNN training optimization.
>
> Q: * in Eq-1/2 is undefined.
> Thanks for pointing out this problem. We have added the definition in the revision.
>
> Q: Confused bold-face in the table.
> Thanks for point out this problem. The boldface just wants to show the best results by FSKD, which may be not the best for all the cases.  We remove the bold-face in the table in our revision.

---

### Official Review · AnonReviewer2 · 2018-11-01
**Surprisingly good model distillation given few samples and non-iterative solution, but practical implications are unclear**

**Rating:** 6
**Confidence:** 4

**Review:**

Model distillation can be tricky and in my own experience can take a lot of samples (albeit unlabeled, so cheaper and more readily available), as well as time to train. This simple trick seems to be doing quite well at training students quickly with few samples. However, it departs from most student-teacher training that find its primary purpose by actually outperforming students trained from scratch (on the full dataset without time constraints). This trick does not outperform this baseline, so its emphasis is entirely on quick and cheap. However, it's unclear to me how often that is actually necessary and I don't think the paper makes a compelling case in this regard. I am borderline on this work and could probably be swayed either way.

Strengths:
- It's a very simple and fast technique. As I will cover in a later bullet point (under weaknesses), the paper does not make it clear why this type of model distillation is that useful (since it doesn't improve the student model over full fine-tuning, unlike most student-teacher work). However, the reason why I do see some potential for this paper is because there might be a use case in quickly being able to adapt a pretrained network. It is very common to start from a pretrained model and then attach a new loss and fine-tune. Under this paradigm, it is harder to make architectural adjustments, since you are starting from a finite set of pretrained models made available by other folks (or accept the cost of re-training one yourself). However, it is unclear how careful one needs to treat the pretrained model if more fine-tuning is going to occur. If for instance you could just remove layers, drop some channels, glue it all together, and then that model would still be reasonable as a pretrained model since the fine-tuning stage could tidy everything up, then this method would not be useful in this situation.
- The fact that least squares solvers can be used at each stage, without the need for a final end-to-end fine-tune is interesting.
- It is good that the paper demonstrates improvements coupled with three separate compression techniques (Li et al., Liu et al., Guo et al.).
- The paper is technically thorough.
- It's good that the method is evaluated on different styles of networks (VGG, ResNet, DenseNet).

Weaknesses:
- Limited application because it only makes the distillation faster and cheaper. The primary goal of student-teacher training in literature is to outperform a student trained from scratch by the wisdom of the teacher. It ties into this notion that networks are grossly over-parameterized, but perhaps that is where the training magic comes from. Student-teacher training acknowledges this and tries to find a way to benefit from the over-parameterized training and still end up with a small model. I think the same motivation is used for work in  low-rank decomposition and many other network compression methods. However, in Table 1 the "full fine-tune" model is actually the clear winner and presented almost as an upper bound here, so the only benefit this paper presents is quick and cheap model distillation, not better models. Because of this, I think this paper needs to spend more time making a case for why this is so important.
- Since this technique doesn't outperform full fine-tuning, the goal of this work is much more focused on pure model compression. This could put emphasis on reducing model size, RAM usage reduction, or FLOPS reduction. The paper focuses on the last one, which is an important one as it correlates fairly well with power (the biggest constraint in most on-device scenarios). However, it would be great if the paper gave a broader comparison with compression technique that may have slightly different focus, such as low-rank decomposition. Size and memory usage could be included as columns in tables like 1, along with a few of these methods.
- Does it work for aggressive compression? The paper presents mostly modest reductions (30-50%). I thin even if accuracy takes a hit, it could still work to various degrees. From what I can see, the biggest reduction is in Table 4, but FSKD is used throughout this table, so there is no comparison for aggressive compression with other techniques.
- The method requires appropriate blocks to line up. If you completely re-design a network, it is not as straightforward as regular student-teacher training. Even the zero-student method requires the same number of channels at certain block ends and it is unclear from the experiments how robust this is. Actually, a bit more analysis into the zero student would be great. For instance, it's very interesting how you randomly initialize (let's say 3x3) kernels, and then the final kernels are actually just linear combinations of these - so, will they look random or will they look fairly good? What if this was done at the initial layer where we can visualize the filters, will they look smooth or not?

Other comments:
- A comparison with "Deep Mutual Learning" might be relevant (Zhang et al.). I think there are also some papers on gradually adjusting neural network architectures (by adding/dropping layers/channels) that are not addressed but seem relevant. I didn't read this recently, but perhaps "Gradual DropIn of Layers to Train Very Deep Neural Networks" could be relevant. There is at least one more like this that I've seen that I can't seem to find now.
- It could be more clear in the tables exactly what cited method is. For instance, in Table 1, does "Fine-tuning" (without FitNet/FSKD) correspond to the work of Li et al. (2016)? I think this should be made more clear, for instance by including a citation in the table for the correct row. Right now, at a glance, it would seem that these results are only comparing against prior work when it compares to FitNet, but as I read further, I understood that's not the case.
- The paper could use a visual aid for explaining pruning/slimming/decoupling.

Minor comments:
- page 4, "due to that too much hyper-parameters"
- page 4, "each of the M term" -> "terms"
- page 6, methods like FitNet provides" -> "provide"

---

> ### Author Response · Authors · 2018-11-26
> **Response to Reviewer2**
>
> Thank you for your review and suggestions, we are happy to address your concerns.
>
> Q: Limited applications
> We agree that teacher-student framework may make student-net has better accuracy than training the student-net from scratch with classification loss (like cross-entropy loss). Our target is fast knowledge distillation from few samples, which has many potential situations for application:
> First, on-device learning to compression when the device has resource constraints which require cheap knowledge distillation solutions.
> Second, when software/hardware vendors want to use knowledge distillation for model compression while the full data is not available due to privacy and confidential issues.
> Third, there is a strict time budget for model optimization so that full training or fine-tuning is not allowed.
> Thank you for your suggestions and we will articulate these reasons in the revision.
>
> Q: Resource usages in the Tables?
> Thanks for your suggestions, we added the resources usages for all our experiments (Table-1~4). Note the parameter pruned ratio is usually higher than the FLOPs reduced ratio.
>
> Q: Aggressive compression case
> In Table1~3, previously we only show the FLOPs reduction ratio, which may make people confused that our compression ratio is relatively low. However, when we add the compression ratio, we could find that in some cases, the compression ratio is also very high (>=85%) not just for Table-4. Nevertheless, we introduce a novel “iteratively pruning and FSKD” procedure similar to that in filter-pruning and network slimming, in Appendix-D. This procedure can produce even larger compression ratio (88%), as shown in the experiment on CIFAR-10.
>
> Q: Visualize the filter before and after FSKD for zero student net?
> The 3*3 kernel is too small to be visualized. However, based on the network decoupling theory, any regular convolution could be decoupled into a sum of depthwise separable convolution blocks, where each block consists of a depthwise conv-layer followed by a pointwise (1*1) conv-layer. The pointwise layer is just a linear combination of channels from the depthwise layer. We then visualize the pointwise conv-layer before SGD, after SGD, after SGD + FSKD for the zero-student case in Appendix-F.
> The experiment shows FSKD improves the smoothness of pointwise convolution tensors.
>
> Q: Cite and comparison with two papers?
> Thanks for pointing us to these two papers. We have cited and compared with them in our revision.
>
> Q: Cite for fine-tuning?
> Yes, the fine-tuning for filter-pruning/network slimming is the same as these two papers did. We have cited them in the revision.
>
> Q: Illustrate the FSKD case for filter-pruning and network decoupling
> We have illustrated these two cases in Figure-5 and Figure-6 in Appendix-C. Please see the revised paper for more details.
>
> Q: Why Q should be squared?
> Squared Q ensures model compression and connectable to the next block (output channel number matches to the input channel number of next block). We have mentioned this in our revision after Theorem-1.

---

### Official Review · AnonReviewer1 · 2018-11-04
**A new formulation of knowledge distillation**

**Rating:** 4
**Confidence:** 4

**Review:**

This paper proposes a framework for few-sample knowledge distillation of convolution neural networks. The basic idea is to fit the output of the student network and that of the teacher network layer-wisely. Such a regression problem is parameterized by a 1x1 point-wise convolution per layer (i.e. minimizing the fitting objective over the parameters of 1x1 convolutions). The author claims such an approach, called FSKD, is much more sample-efficient than previous works on knowledge distillation. Besides, it is also fast to finish the alignment procedure as the number of parameters is smaller than that in previous works. The sample efficiency is confirmed in the experiments on CIFAR-10, CIFAR-100 and ImageNet with various pruning techniques. In particular, FSKD outperforms the FitNet and fine-tuning by non-trivial margins if only small amount of samples are provided (e.g. 100).

Here are some comments:

1. What exactly does “absorb” mean? Is it formally defined in the paper?

2. “we do not optimize this loss all together using SGD due to that too much hyper-parameters need tuning in SGD”. I don’t understand (1) why does SGD require “too much” hyper-parameters tuning and (2) if not SGD, what algorithm do you use?

3. According to the illustration in 3.3, the algorithm looks like a coordinate decent that optimizing L over one Q_j at a time, with the rest fixed. However, the sentence “until we reach the last block in the student-net” means the algorithm only runs one iteration, which I suspect might not be sufficient to converge.

4. It is also confusing to use the notation SGD+FSKD v.s. FitNet+FSKD, as it seems SGD and FitNet are referring to the same type of terminology. However, SGD is an algorithm, while FitNet is an approach for neural network distillation.

5. While I understand the training of student network with FSKD should be faster because the 1x1 convolution has fewer parameters to optimize, why is it also sample-efficient?

6. I assume the Top-1 accuracies of teacher networks in Figure 4 are the same as table 2 and 3, i.e. 93.38% and 72.08% for CIFAR-10 and CIFAR-100 respectively. Then the student networks have much worse performance (~85% for CIFAR-10 and ~48% for CIFAR-100) than the teachers. So does it mean FSKD is not good for redesigned student networks?

7. While most of the experiments are on CIFAR10 and CIFAR100, the abstract and conclusion only mention the results of ImageNet. Why?

---

> ### Author Response · Authors · 2018-11-26
> **Response to Reviewer1**
>
> Thanks for the valuable comments and suggestions. We give detailed responses to each item below.
>
> 1. Meaning of “absorb”
> We add a 1x1 conv-layer Q \in R^{n_o’ * n_o * 1 * 1} to student-net after the conv-layer W\in R^{n_o *n_i * k* k} before non-linear layer, “absorb” here means Q can be merged into W to obtain a new conv-layer W’ \in R^{n_o’ * n_i *k *k}. If Q is squared (n_o’=n_o), then W’ \in R^{n_o * n_i *k*k} has the same size as W. Previously we put this information at appendix-A, and now we revise the description of Theorem-1 to include the information.
>
> 2&3: why not SGD, why use one-step block coordinate descent?
> Yes, what we use is in fact one-step block coordinate descent (BCD) algorithm. We add a description of our BCD algorithm in the appendix-B, also include experiment comparison to FSKD-SGD (total loss optimization with SGD on all added 1x1 convs’ parameters together) and FSKD-BCD in the experiments on filter-pruning. The experiments show that FSKD-BCD clearly outperforms FSKD-SGD in all cases. The advantages of the BCD algorithm are also listed in the revision. One major reason is that BCD each time handles few parameters (one block) which can be solved with limited samples, while SGD always takes all added 1x1 convs’ parameters into consideration, thus requires more data in the optimization.
> Our experiments do not show benefit from more iterations of BCD.
> This may be due to the fact that the added 1x1 conv-layer is before non-linear activations so that one-step linear estimation is accurate enough to get exact minimization. Hong et al [1] show BCD can reach sublinear convergence when each block is exactly minimized, which is consistent with our experiments. We also add Fig3b to illustrate the accuracy improvement along with block alignment sequentially.
>
> [1] Hong, Mingyi, et al. "Iteration complexity analysis of block coordinate descent methods." Mathematical Programming 163.1-2 (2017): 85-114.
>
> 4. Confusing on SGD+FSKD and FitNet+FSKD
> We denote SGD as optimization without using teacher-net info. For the zero-net experiment, SGD+FSKD first uses SGD to initialize the student-net on few samples without using teacher-net info, then uses FSKD to further improve the performance with teacher-net info. While the Fitnet+FSKD first uses FitNet to initialize the student-net on few-samples with teacher-net guidance, then uses FSKD to further improve the performance with teacher-net info from our FSKD perspective. We clarify this in our revision.
>
> 5. Why FSKD is sample efficient?
> As is known, fewer parameters tend to require fewer samples for estimation. The BCD algorithm considers each block separately, thus there are much fewer parameters in each block, so that we could use much fewer samples for the block-level estimation. Our experiments also verify this point when comparing FSKD-BCD to FSKD-SGD in Figure-2, especially when samples < 100.
>
> 6. Not good on redesigned student-net?
> Yes, in Figure-4, the student-net accuracy is about 83% on CIFAR-10 and about 47% on CIFAR-100. We should emphasize that this result is obtained with a very limited number of training samples and without data augmentation. If data augmentation is enabled, about 5% accuracy improvement could be achieved with FSKD. We design this experiments to demonstrate the effectiveness of FSKD over FitNet and SGD on the few-sample settings, and we do not compare this result with full-data training.
>
> 7. Only mention results on ImageNet in abstract and conclusion.
> We have revised our abstract and conclusion accordingly, even though the results on ImageNet sounds more significant to us.

---

> > ### Comment · AnonReviewer1 · 2018-12-09
> > **Rebuttal read. Concerns remain.**
> >
> > - Hong et al [1] focuses on BCD for convex optimization problem, which is very different from the proposed formulation. So I think its theoretical result has nothing to do with the your method.
> >
> > - “The BCD algorithm considers each block separately, thus there are much fewer parameters in each block, so that we could use much fewer samples for the block-level estimation.”
> >
> > This explanation sounds weird to me. According to this logic, we can always set the number of parameters in each block to be 1. Then it will become even more sample efficient.
> >
> > - “83% on CIFAR-10 and about 47% on CIFAR-100”
> >
> > Those are really bad performances, given that the teacher network can achieve 93.38% and 72.08% for CIFAR-10 and CIFAR-100. Usually, a distillation network can achieve similar performance as the teacher network (see [1]). Then I confirm my conclusion that the proposed technique is far from being used.
> >
> > Therefore, I will keep my rating.
> >
> > [1] Distilling the Knowledge in a Neural Network. Geoffrey Hinton, Oriol Vinyals, Jeff Dean. NIPS 2015.

---

> > > ### Author Response · Authors · 2018-12-09
> > > **further response**
> > >
> > > Thanks for the valuable comments and suggestions. Below further response your concerns.
> > >
> > > ### Hong et al focuses on convex optimization problem
> > > We agree that CNN optimization problem ins non-convex. However, our problem is not a standard CNN optimization problem. The loss function in eq(2) contains multiple disjoint blocks without non-linear activation function in each block (only between two blocks), while all the other network parts are fixed.
> > > Even considering the non-linear activation function after each block (Q), the loss is piece-wise linear.
> > > With a prox-linear surrogate, the global convergence can be found by minimizing the prox-linear  surrogate as supposed by [1].
> > > [1] Xu, Yangyang, and Wotao Yin. "A globally convergent algorithm for nonconvex optimization based on block coordinate update." Journal of Scientific Computing 72, no. 2 (2017): 700-734.
> > >
> > > ### BCD requires few samples, why not setting parameter to 1?
> > > Sorry for the inaccurate descriptions. For our cases, the blocks are multiple disjoint blocks, it is not like those of coordinate descent in which variables are correlated to each other. Due to the disjoint properties, we can use relatively fewer samples to estimate parameters in each block.
> > >
> > > ### performance on zero-student-net not comparable to teacher-net.
> > > We should  emphasize that there are some new attempts which try to realize knowledge distillation with few samples, while all of them do not show good performance, including [2] and [3] on MNIST, [4] on CIFAR-10 and MNIST.
> > > We should emphasize again that we have only use quite a few samples without data augmentation for this study, which still achieves much better accuracy than SGD and FitNet. With data augmentation, the accuracy can be improved about 5%. This is acceptable considering such a few original samples used.
> > > Furthermore, we emphasize the great benefits of the proposed framework on student-net by
> > > extremely decomposed/pruned from teacher-net. We specially emphasize the usage of this setting at the second but last paragraph.
> > >
> > > [2] Akisato Kimura, Zoubin Ghahramani, Koh Takeuchi, et al. Few-shot learning of neural networks from scratch by pseudo example optimization.  (big gap on MNIST SOTA performance with few samples).
> > > [3] Raphael Gontijo Lopes, Stefano Fenu, Thad Starner, et al. Data-free knowledge distillation for deep neural networks. arXiv preprint arXiv:1710.07535, 2017.
> > > [4] dataset distillation, ICLR 2019 submission.

---

### Public Comment · (anonymous) · 2018-10-27
**Question about the optimize of loss**

I'm trying to reproduce your results in Sec. 4, but had a question about the the optimize of loss in Sec. 3.3 Algorithm 1:
1. not use SGD optimize the loss, instead by Algorithm 1, but how does it work well, I can't understand here. What more can you describe?.



Thanks!

---

> ### Author Response · Authors · 2018-10-28
> **Clarification on the optimization process**
>
> Thanks for your comment!
>
> Yes, our problem can be optimized using SGD with loss function defined in Eq(2) . However, we did not use it to report results in the paper. Instead, we estimate the 1x1 conv-layer parameter Q by solving the least squared problem layer by layer sequentially.
>
> To be more specific, given randomly selected few-samples, we forward the data to the alignment point of the first block in both student network and teacher network, and obtain the feature map responses at this point. Suppose the teacher network response is X^t, and the student-net response is X^s, we obtain Q using X^s and X^t with Eq(1). Then based on our Theorem-1, we absorb the 1x1 conv defined by Q into previous conv-layer. After that, we move to the alignment point of the next block, and repeat this procedure until we reach to the final alignment point. This simple solution works well since the alignment point is before non-linear activation function. Figure-3 shows the block-level correlation before and after alignment between teacher and student, which also demonstrate this effectiveness of this linear approximation.
>
> We use this procedure instead of the SGD based optimization due to the following reasons.
>
> (1)  We in fact implement the SGD based solution on the filter-pruning and slimming experiments. But we did not find noticeable results difference between these two solutions.
>
> (2)  SGD requires tuning several hyper parameters, while our simple solution is hyper-parameter free. We find it is relatively difficult to tune SGD based solution on the network decoupling case due to multi-branch network structures. There are no advantages on time budget over the proposed simple solution.
>
> (3)  Our experiments demonstrates the proposed simple solution works pretty well for the cases, and we also have some figures which illustrates steady accuracy improvement during block-by-block alignment. And we will include that in the revision.
>
> We will also make our source code available in the near future.

---

### Public Comment · (anonymous) · 2018-10-28
**About "Q is degarded to the matrix form"**

I don't know "Q is degarded to the matrix form". Could you tell what is the specific operation here? Any other references?

Thanks！

---

> ### Author Response · Authors · 2018-10-28
> **Clarification on Q**
>
> Thanks for your comment!
>
> We use Q to denote the parameters of the added 1x1 conv-layer, the size is nxmx1x1, where n is the input channel number, m is the output channel number, 1x1 is the kernel size. The 4D tensor is then degraded to a matrix with size nxm. Q acts as a linear combination of the input and output channels. For more information about the 1x1 convolution, please refer to [1].
> To be more specific, suppose Q_{ij} is the element of the matrix  Q at i-th row and j-th column, it reflects the combination coefficient between input channel i and output channel j.  This is how we represent Q as a matrix.
>
> [1] Min Lin, Qiang Chen, and Shuicheng Yan. "Network in network." arXiv preprint arXiv:1312.4400 (2013).

---

### Comment · AnonReviewer2 · 2018-10-30
**Clarification on non-square Q**

I'm a bit confused by the fact that in sec 3.3, Q is said to be square (satisfy c4). Why is this always satisfied here, because I thought the whole point was that the student might have fewer channels than the teacher. I guess there are a couple of different situations to consider (such as reducing channels, or reducing layers). In the former, the way I understood the training was as following: We start with after the first layer (let's say teacher output has 128 channels and student 64). We construct a Q with shape (64, 128, 1, 1) and and assume the first layer in the student is the same as the teacher. We solve it and absorb Q into this weight for the student. Note, at this point if the student also had 128 channels output, there would be no need to do solve for a Q. Next, we look at the activations of the teacher after the second layer (let's say it's still 128). This is where we run into the first issue I'm not sure how to address, since the weights of the teacher layer is no longer compatible with the student, so we cannot use it anymore. We would have to first absorb the inverse of the previous Q into this weight, to get us back to 128 channels going into that layers. I guess that's when you don't copy weights from the teacher and instead initialize randomly (zero student).

Anyway, a bit more clarity on when you can re-use original teacher weights and when you have to randomly initialize - as well as when Q is square and when Q is non-square. Thanks!

---

> ### Author Response · Authors · 2018-10-31
> **Three detailed cases how Q is defined**
>
> Thanks for your comments!
>
> First we clarify the initialization problem. In section 3.3, we conduct two sets of experiments. The first set obtains student-net from compressing teacher-net, including filter pruning, network slimming, and network decoupling. The second set fully redesigned the student-net with a different structure from the teacher-net and random initialized the parameters (i.e., zero net in the paper).
> For the first set of experiments, the student-net already has an initialization from original teacher-net's weights.
> For the second set of experiment, we start the student-net with random weights, then use SGD to initialize the student-net using few samples before adopting our FSKD.
>
> Second, Q is required to be squared in condition-4 (c4) due to two reasons.
> (1) Q must be squared to make current layer and next layer connectable.
> (2) If Q is not squared, it will decrease the compression effect after absorbing Q into previous layer.
> Let's give an example to explain that.
> Suppose current conv-layer is 64*64*k^2 (64 channels in and 64 out, k^2 is the spatial kernel size), and next layer is 64*128*k^2 (64 channels in and 128 out).
> If we set Q being 64*128*1^2, after absorbing, the current layer will be 64*128*k^2, which can't connect to next layer (with size 64*128*k^2).
> Besides, it also increases the parameter number and computing cost quite a lot for current layer.
>
> Third, we list the 3 cases how Q is defined.
> (1) For the fully redesigned student network (zero net), we ensure that the corresponding block-level output channels are matched between teacher and student.  If the block output channel number is n, then Q is a matrix with size n*n.
> (2) For the network decoupling case, the regular convolution and depthwise separable block has the same number of output channels, it is also straightforward to define the size of Q.
> (3) For the  pruning and slimming cases, there are two different sub-cases.
> 3a) When there are multi-layers within an alignment block, the student-net may either have less layers or smaller intermediate channels in the block comparing to the teacher-net, but still keep both teacher and student have the same number of output channels for that corresponding block. For instance, the teacher-net has a block with 2 conv-layers (64*64*k^2 followed by 64*128*k^2), the student-net may just have less conv-layers (1 here) in the block as (64*128*k^2). Or the student-net may have smaller number of internal channels in the block as (64*32*k^2, 32*128*k^2), here 32 is the number of  output channels for the first layer and number of input channel for the second layer.  For both examples, Q should be 128*128, so that it could be absorbed into previous conv-layer.
>
> 3b) When we do per-layer alignment, suppose the layer of the teacher-net is 64*128*k^2. After pruning, we have the corresponding layer in student-net as 64*64*k^2. The #channels between teacher and student are different here. We split the 128 channels of teacher-net into two parts, pruned 64 channels and unpruned 64 channels. We make a linear regression between the student-net and the unpruned 64 channels of teacher-net, so that we defined the matrix Q with size 64*64.  For the first layer, the alignment matrix Q is an identical matrix since the unpruned part of teacher-net is copied to the student-net. However, when moving to next layer, for teacher-net, the input information will come from all the 128 channels of previous conv-layer. But for the student-net, the input information will come from only the 64 channels of previous conv-layer.  There is obvious information loss, so that we estimate Q to alleviate this information loss.
>
> We will clarify this in our revision with text explanation and figure illustrations.

---

### Public Comment · (anonymous) · 2018-11-13
**absorb convolution?**

What does it mean with absorb 1x1 conv layer to previous conv layer?
I think absorb is too ambiguous for a word
Copy weights to previous conv layer?
Stack trained 1x1 conv on top of previous layer but below ReLU?

Need better explainability, some figure to visualize would help

---

> ### Author Response · Authors · 2018-11-26
> **absorbing is defined in the revision**
>
> Thanks for the question. We add definition in our revision.
> Please see the text we answer to Reviewer-1 for more explanation.
> We also visualize how FSKD works for filter pruning/network slimming and the network decoupling cases.
> Please check our Appendix-C for more details.

---

### Author Response · Authors · 2018-11-26
**Revision list for the updated version**

Thanks to the valuable suggestions and comments from reviewers and anonymous commenter, we carefully revise our paper accordingly. Here we list major revision points below.
1)      Add reasons why knowledge distillation from few samples is important.
2)      Revise abstract/conclusion to replace ImageNet description with general descriptions.
3)      Add comparison to two related papers in related work
4)      Revise Theorem-1 to include the definition of “absorb”, and text after Corollary-1 why Q should be squared.
5)      Revise text after Eq-2 to clearly present that our algorithm is based on block coordinate descent (BCD), and the advantages of BCD.
6)      Fig2 add experimental results comparison between FSKD-BCD and FSKD-SGD.
7)      Add Fig3b to show the BCD accuracy improvement along with block alignment.
8)      Table1~4, add columns for parameters number and pruned.
9)      Zero-net adds more descriptions and analysis.
10)    Appendix-B: the BCD algorithm and one experiment to show the impact of iteration number.
11)    Appendix-C: Fig5 and Fig6 for illustrations of how FSKD works on filter pruning and network decoupling.
12)    Appendix-D: iterative pruning and FSKD to achieve extremely pruned network and one experiment (scheme-C) on VGG-16 on CIFAR-10.
13)    Appendix-E: verification of the hypothesis pointwise convolution is more critical for performance than depthwise convolution.
14)    Appendix-F: Filter visualization on zero-student-net before SGD, after SGD, and after SGD+FSKD.

---

### Meta-Review · Area_Chair1 · 2018-12-13
**limited novelty, unclear motivation**

**Confidence:** 5
**Recommendation:** Reject

**Metareview:**

The paper considers the problem of knowledge distillation from a few samples. The proposed solution is to align feature representations of the student network with the teacher by adding 1x1 convolutions to each student block, and learning only the parameters of those layers. As noted by Reviewers 1 and 2, the performance of the proposed method is rather poor in absolute terms, and the use case considered (distillation from a few samples) is not motivated well enough. Reviewers also note the method is quite simplistic and incremental.